# Sensitivity of Nutrition Indicators to Measure the Impact of a Multi-Sectoral Intervention: Cross-Sectional, Household, and Individual Level Analysis

**DOI:** 10.3390/ijerph17093121

**Published:** 2020-04-30

**Authors:** Anastasia Marshak, Helen Young, Anne Radday, Elena N. Naumova

**Affiliations:** 1Feinstein International Center, Tufts University, Boston, MA 02111, USA; Helen.young@tufts.edu (H.Y.); Anne.radday@tufts.edu (A.R.); 2Friedman School of Nutrition Science and Policy, Tufts University, Boston, MA 02111, USA; elena.naumova@tufts.edu

**Keywords:** nutrition, early childhood, multi-sectoral programming, mixed-effects model, Chad, analytical approach

## Abstract

Interventions tackling multiple drivers of child malnutrition have potential, yet the evidence is limited and draws on different analysis and nutrition outcomes, reducing comparability. To better understand the advantages and disadvantages of three different analytical approaches on seven common nutrition indicators, we use panel data (2012, 2014, 2015) on 1420 households from a randomized control study of a multi-sectoral intervention in Chad. We compare program impact using three types of analysis: a cross-sectional analysis of non-matched children; a panel analysis on longitudinal outcomes following the worst-off child in the household; and a panel analysis on longitudinal outcomes of matched children. We find that the sensitivity of the nutrition outcomes to program impact increases with each subsequent analytical approach, despite the reduction in sample size, as the analysis is able to control for more non-measured child and household characteristics. In the matched child panel analysis, the odds of a child being severely wasted were 76% lower (CI: 0.59–0.86, *p* = 0.001), the odds of being underweight were 33% lower (CI: 0.15–0.48, *p* = 0.012), and weight-for-height z-score was 0.19 standard deviations higher (CI: 0.09–0.28, *p* = 0.022) in the treatment versus control group. The study provides evidence for multi-sectoral interventions to tackle acute malnutrition and recommends the best practice analytical approach.

## 1. Introduction

In 2015, the international community formally adapted an ambitious new sustainable development agenda. A portion of the Sustainable Development Goals (SDGs) focuses on achieving nutrition targets for children under five years of age, specifically, by 2025, to reduce wasting or weight-for-height z-score (WHZ) below minus two standard deviations to 5% and reduce stunting or height-for-age z-score (HAZ) below minus two standard deviations by 40%. By 2030, the SDG target is to end all forms of malnutrition [1]. However, not all countries are at pace to meet these targets. West Africa witnessed an increase in stunting from 2000 to 2015 [2]. Furthermore, specifically Chad continues to rank second to last on the Global Hunger Index (which combines wasting, stunting, undernourishment, and under-five mortality) and has shown no improvement since 2000 [3].

A recent approach to tackle child malnutrition in all of its forms, adapted by nutrition stakeholders in Chad, established a national multi-sectoral plan that aims to address all possible drivers of child wasting and stunting, from infant and young child feeding, to food security, water, health access, and livelihood support [4,5,6]. However, to date, despite the global and national push for more multi-sectoral programming (the Scaling Up Nutrition (SUN) Movement, the Common Results Framework, USAID Multi-sectoral Nutrition Strategy), the evidence remains limited [7,8] and non-existent in Chad. The evidence base that does exist relies on a variety of different analytical approaches and nutrition outcomes to establish whether these programs had an impact, making an overall assessment challenging. For example, a randomized control trial in Niger compared hazard ratios for the incidence of severe wasting (WHZ less than minus three standard deviations) and moderate wasting (WHZ between minus three and minus two standard deviations) between the different treatment arms over five months of intervention [9]; a study across nine sub-Saharan African countries looked at the prevalence of stunting, HAZ, underweight (weight-for-age z-score (WAZ) less than minus two standard deviations), and wasting across a baseline and endline sample, comparing similar children across the three-year time period using propensity score matching [10]; a randomized control trial in Bangladesh compared the prevalence of stunting and wasting over seven years of intervention using paired and unpaired t-tests between baseline and endline [11]; and a randomized control trial in Haiti looked at stunting, HAZ, underweight, WAZ, wasting, and WHZ between baseline and endline across three years of intervention using both paired t-test comparisons at the endline and random-effects models across the two time periods. Thus, while all these studies relied on the gold standard of randomized control trials, the outcome indicators and analytical approaches varied widely.

To the authors’ knowledge, few papers offer the comparison of different analytical approaches on the same dataset in relation to different nutrition outcomes. A study in northern Nigeria came close by comparing the prevalence of wasting from a mixed panel and cross-sectional data collection (over 10 months) using a linear mixed-effects model. The study found that repeated monthly measures on wasting in the same cluster-village, as opposed to random selection of new villages, are prone to observational bias with the sentinel sites showing progressive improvement and hence, deviating from that of the community they are meant to represent [12].

To better help understand the value of different analytical approaches and nutrition outcome sensitivity when it comes to assessing the effect of an intervention, we explore data from a four-year multi-sectoral program—Community Resilience to Acute Malnutrition (CRAM)—carried out in collaboration by researchers at Tufts University and the non-governmental organization (NGO) Concern Worldwide. In 2012, CRAM integrated three program sectors: livelihoods; water, sanitation, and hygiene (WASH); and nutrition and health, and thus, is broadly defined as a multi-sectoral program, implemented in the Kimiti Department of Chad. Some specific program examples include providing seeds and training for dry season market gardening, construction of wells and latrines, establishing mother support groups for pregnant women and those with children under five, and messaging around exclusive and complementary breastfeeding [13,14]. To test program impact on key inputs, outputs, and outcomes, a randomized control impact evaluation was put in place by Tufts researchers, following the same households from 2012 to 2015 [15]. After three years of CRAM programming, impact was observed on some key inputs (increased use of wells and latrines), outputs (increased use and knowledge of hygiene behaviors across the water chain, increased knowledge around hand-washing), and outcomes (increased weight-for-height z-score) [14].

Given the rich data collected over the course of CRAM program, we aim to take advantage of all available findings to improve the methodology of a multi-sectoral program evaluation. Thus, as a best practice for the analytical stage of any investigation, we aim to explore the pros and cons of different models that can be applied to the collected data to learn not only about what the study aimed to achieve, but also unexpected and new results. Thus, using the CRAM data, we explore the sensitivity of seven key nutrition outcome indicators (wasting, severe wasting, WHZ, underweight, WAZ, stunting, and HAZ) that are frequently and often interchangeably used across similar studies, four of which directly relate to the SDGs, to establish the impact of multi-sectoral programs. Severe wasting was included among the seven indicators because it is the measure used by health centers to determine who receives malnutrition treatment, which is part of the multi-sectoral intervention package being evaluated. We applied three different types of analysis across the same dataset: a traditional cross-sectional analysis based on outcomes for non-matched child records for all three years of data; a panel analysis applied to longitudinal outcomes at the household level, focusing on the outcomes of the worst-off and best-off child in the household from a nutritional perspective; and, a longitudinal analysis applied to outcomes of matched children over time. By identifying the analytical approach most suited for determining program impact, we can improve future study design, harmonizing across evaluations for better comparability and growing the evidence base for multi-sectoral approaches to addressing child malnutrition.

## 2. Materials and Methods

The study was conducted in the Kimiti Department of Chad. The sampling frame was pulled from a household list drawn up by the Concern team as part of their humanitarian blanket feeding and general ration program in 2010. The Concern team identified households according to the lead female, to ensure that polygamous households were equitably targeted and to distribute the general ration to the female caretaker. These women served as the survey respondents, as we believed that they would have a more accurate perspective of household dynamics related to children, food, and health. Due to the polygamous nature of the context, it is possible that some female respondents shared the same husband. Oral consent was required for participation in the survey and all instruments were approved by the Tufts Internal Review Board (IRB). The trial was registered with clinicaltrials.gov, reference number NCT03454100.

Prior to the blanket food aid program, the Concern team conducted a participatory wealth ranking in the recipient village. Concern worked with each community to categorize households by their comparative status in terms of livestock ownership, income, and livelihoods (indicators self-selected by the community as proxies for wealth). Following the categorization exercise with the communities, Concern rated each household on a scale from A through D, with A being the wealthiest and D being the poorest category of households. Given that the multi-sectoral intervention only targeted households in the bottom three wealth groups (B, C, and D), random selection within each village for the impact evaluation was also done only from group B, C, and D.

Given a power of 0.80, a significance level of 0.05, a minimum effect size of 0.22, and an intra-class correlation of 0.06, the study required a total of 1400 households clustered in 70 villages. As one village refused to participate, the total number of clusters amounted to 69. In the end, 1420 households were selected, providing data on approximately 1200 children split evenly between the treatment and control group (Appendix A). To meet this requirement and keep clusters of equal size, only villages/clusters that had 20 households or more with a household B, C, or D wealth ranking were selected. Villages were randomly assigned to the treatment and control group. The baseline was done in November/December 2012, the midline in November/December 2014, and the endline in November/December 2015. Due to mortality, migration, and relocation, approximately 11% of the baseline households were not resurveyed in the endline. While this attrition rate is not ideal, household attrition was not correlated to intervention.

Prior to the analysis, we cleaned and examined the household and child data, including the anthropometric data, using Stata/SE 13.1. After transforming the anthropometric data into z-scores, we removed unrealistic values (z-score greater than negative or positive five for weight-for-height and z-scores greater than negative or positive six for height-for-age and weight-for-age) which accounted for 113 (<3%) data points over the full duration of the study. To understand program impact on child nutritional status and compare different approaches to analysis, we ran three different types of analysis. First, we took advantage of the full available data set as if it was three cross-sectional surveys, and thus, treating every child level observation as independent across time. Second, we exploited the original design of the project, and carried out a panel analysis on the households that were tracked over time. Because approximately 50% of households had more than one child (Appendix A), we used the nutritional outcome for the worst-off child (for example: lowest WHZ, or the child severely wasted, etc.,), average child, and best-off child in the household and conducted three separate analyses. This approach allowed us to use child nutrition as our outcome variable while treating the household as the main unit of analysis, and hence, controlling for omitted household level characteristics, but not child level. Third, we identified children that were measured multiple times to carry out a child-level longitudinal panel analysis. To match the same child across time, we used the following criteria:Child had to be in the same household over time.Child had to be of the same gender over time.Child had to be within 12 or 24 months (depending on the difference in time between data collection) with a margin of error of plus or minus six months over time. Capturing the true age of a child is notoriously difficult in settings like eastern Chad, due to lack of birth certificates and other documentation. Hence, enumerators use seasons and holidays to place the age of the child and there is likely a wide margin of error.

On average, 38% of all children for which we had anthropometric data were matched over time. There was no difference with a *p*-value < 0.05 in nutrition indicators across children that were matched and those that were not matched at each time period (Appendix A). Overall, for each additional approach, the trade-off was sample size for efficiency (see sample size by analysis in Appendix A).

Comparison between children in the treatment and control group at each individual point in time (2012, 2014, and 2015) was done using simple t-tests, controlling for sample design and applying population weights. The comparison was run on the seven outcomes’ variables (Table 1) for all children under the age of five: wasting, severe wasting, WHZ, stunting, HAZ, underweight, and WAZ.

To take advantage of the multi-year and nested nature of the data, we used a mixed effects regression model adapted for both continuous and binary outcomes (Model 1). The mixed effects regression was applied to both the matched household sample for the nutritionally best-off and worst-off child in the household analysis (Model 2) and the matched child sample analysis (Model 3). For each model, we assessed the crude effect, meaning just the impact of the intervention over time, and the adjusted effect, where we controlled for some child (age and sex) and household level (number of children in the household and if a child died in the past year) characteristics. The following equations were used for Model 1 (non-matched child), Model 2 (matched household), and Model 3 (matched child) respectively:(1)$$Y_{i}=\beta_{0}+\beta_{1}I_{i}+\beta_{2}t_{i}+\beta_{3}I_{i}\times t_{i}+\beta_{k}X_{ki}+\gamma_{k}Z_{kj}+\varepsilon_{i}$$
(2)$$Y_{it}=\beta_{0}+\beta_{1}I_{i}+\beta_{2}t_{i}+\beta_{3}I_{i}\times t_{i}+\beta_{k}X_{kit}+\gamma_{k}Z_{kjt}+\alpha_{i}+\varepsilon_{it}$$
(3)$$Y_{ijt}=\beta_{0}+\beta_{1}I_{ij}+\beta_{2}t_{ij}+\beta_{3}I_{ij}\times t_{ij}+\beta_{k}X_{kijt}+\gamma_{k}Z_{kjt}+\alpha_{i}+\varepsilon_{ijt}$$
where:$Y_{ijt}$ is the outcome variable for *i*-child from *j*-household at *t*-time (*t* = 0, 2, 3 for 2012, 2014, and 2015). In Equation (2), *j* equals *i* because we only have one observation per household (nutritionally worst-off or best-off child in the household);$I_{ij}$ represents intervention (treatment vs. control) for *i*-child from *j*-household;$t_{ij}$ represents time for *i*-child from *j*-household;$I_{ij}\times t_{ij}$ represents the interaction term between the intervention and time for *i*-child from *j*-household;$X_{kijt}$ represents the vector of *k*-control child-level variables for *i*-child from *j*-household at *t*-time;$Z_{kjt}$ represents the vector of *k*-control household-level variables for *j*-household at *t*-time;$\beta_{0}$ is the intercept;$\beta_{1}$ is the impact of the intervention in 2012;$\beta_{2}$ is the impact of time in the control group;$\beta_{3}$ is the impact of the intervention over time;$\gamma_{k}$ is a vector of household-specific intercepts;$\alpha_{i}$ is the error term for *i*-child;$\varepsilon_{ijt}$ is the error term for *i*-child from *j*-household at *t*-time.

Equations (1)–(3) (for continuous outcomes) were then adapted to binary outcomes. For example, for Model 1 the logit form is formulated as:(4)$$\ln(P_{i})=\ln\left(\frac{P_{i}}{1-P_{i}}\right)=\beta_{0}+\beta_{1}I_{i}+\beta_{2}t_{i}+\beta_{3}I_{i}\times t_{i}+\beta_{k}X_{ki}+\gamma_{k}Z_{kj\;}+\varepsilon_{i}$$
where *P_i_* is the probability of *i*-child being wasted, stunted, or/and underweight.

To calculate the effect size and 95% confidence interval (CI) at *t*-time into an easily interpretable value, indicating the % increase or decrease in the odds that a child is malnourished in the treatment group relative to the control group at t-time, we used the following equations:(5)$$e^{\left(\left(\beta_{1}+\;\beta_{3}*t\right)\;\pm \;\;1.96*\sigma_{\beta_{3}}\right)}$$

For continuous outcomes, we calculate the size of the impact in z-score standard deviations where $\sigma_{\beta_{3}}$ is the standard error of the interaction term (Equation (6)):(6)$$(\beta_{1}+\beta_{3}\times t)\;\pm \;1.96\times \sigma_{\beta_{3}}$$

## 3. Results

At the endline (2015), using simple *t*-tests, accounting for the design effect and applying population weights, an impact with a *p*-value < 0.05 was observed on three outcome indicators: WHZ was 0.29 standard deviations higher in children living in the treatment villages (mean: −0.80; CI: −0.98 to −0.63; *p*-value: 0.015) compared to the control villages (mean: −1.09; CI: −1.24; −0.94); % of underweight children was 8.3 percentage points lower in the treatment group (mean: 28.7%; CI: 23.8 to 37; *p*-value: 0.011) compared to control (mean: 37%; CI: 33.1 to 40.9); and WAZ was 0.20 standard deviations higher in the treatment group (mean: −1.46; CI: −1.59; −1.33; *p*-value: 0.042) compared to control (mean: 38.1; CI: 32.3; 43.9) (Figure 1 and Appendix A).

Results from Model 1 (Table 2 for adjusted model results and Appendix A for crude model results), combining all three time periods of unmatched child data, show that the intervention had an impact on WHZ (coefficient: 0.06; CI: 0.00 to 0.12; *p*-value = 0.039). The data show that WHZ was 0.19 standard deviations higher (CI: 0.13 to 0.25) in the treatment group as opposed to the control group. Girls were better off for six of the seven outcome variables with a *p*-value < 0.05.

Prior to running the mixed effects models (Model 2), we explored whether there was a difference in mean value across treatment when using the nutrition outcomes for the nutritionally worst-off, best-off, and average child in the household. We consistently found that the nutritionally best-off child did have far better outcomes compared to the worst-off child, and the average child fell somewhere in the middle (Figure 2 for severe wasting and WHZ; and Appendix A).

In the mixed effects regression on matched households (Model 2), the impact of the intervention was observed on severe wasting (coefficient −0.63; CI: −0.97 to −0.29; *p*-value < 0.00) and WHZ (coefficient: 0.08; CI: 0.01 to 0.14; *p*-value: 0.01) for the worst-off child (Figure 2 and Table 2), expanding on the results seen in the non-matched analysis. The data show that the odds of a child being severely wasted in 2015 were 65% lower (CI: 0.50 to 0.75) and WHZ was 0.18 standard deviations higher (CI: 0.12 to 0.24) in the treatment group as opposed to the control group. When limiting the analysis to only households who had two or more children (Appendix A), the impact is smaller (51% odds (CI: 0.25 to 0.6) of being severe wasting in treatment as opposed to control group in 2015) and only observed on severe wasting (coefficient: 0.61; CI: −1.04; −0.19; *p*-value: 0.005). When we carry out this analysis on the nutritionally best-off child in the household in households with two or more children, there is no evidence for an impact of the intervention (Appendix A).

Finally, we ran a mixed effects model on the child data matched across time (Model 3) (Table 2). Despite more than one-half reduction in the sample size, the matched child analysis shows the impact of the intervention on severe wasting (coefficient: −0.90; CI: −1.44 to −0.37; *p*-value: 0.001), WHZ (coefficient: 0.10; CI: 0.01 to 0.20; *p*-value: 0.022), and underweight (coefficient: −0.30; CI: −0.54; −0.06; *p*-value: 0.012). In 2015, the odds of a child being severely wasted were 76% lower (CI: 0.59 to 0.86) and the odds of being underweight were 33% lower (CI: 0.15 to 0.48) in the treatment group compared to the control group. WHZ was 0.19 standard deviations higher (CI: 0.09 to 0.28) in the treatment group compared to the control group by the end of the impact evaluation. Girls were better off on five out of the seven nutrition outcome indicators.

Combining all the models together (Figure 3), we can more clearly compare the results based on each individual type of analysis. Figure 3 illustrates the added benefit of matched child analysis, showing results with a *p*-value < 0.05 across three of the outcome variables as opposed to two with the matched household analysis and only one with the cross-sectional analysis, despite having less than one-half the sample size.

## 4. Discussion

The design of the CRAM study and subsequent analysis offers insight for future impact studies aiming to address child malnutrition in all its forms and to meet the SDG targets. Previous studies attempting to discern the impact of a multi-dimensional program on different nutrition indicators, were primarily cross-sectional [11,16], prospective observational [10], or prospective intervention studies [9], which limits the analysis to comparing across households rather than within households over time. In addition, the above-mentioned studies used a variety of different outcome indicators, again limiting comparability. Using the CRAM data, we were able to contrast seven commonly used indicators and assess their value by comparing across cross-sectional surveys (Model 1), matched households (Model 2) and matched children (Model 3). While each model has its own individual advantages, the findings clearly illustrate that as the model can control for more non-measured characteristics, we find more nutrition outcomes are sensitive to program impact despite a drop in the sample size.

Model 1 utilizes all available child data with anthropometry across the three years, and thus, has the largest sample size. This approach treats every year of data collection as a cross-sectional survey, and hence, is reminiscent of the most common and simplest evaluation design, but only yields observable impact on WHZ. The second model draws from a smaller dataset on matched households, thus allowing the model to control for heterogeneity in household characteristics that might not necessarily be captured by available variables or are simply unobservable predictors, removing omitted variable bias, expanding our ability to observe impact on WHZ and severe wasting. The third model, relying on matched child data, has a sample size less than half of the first model, but has the added benefit for controlling for not only household level omitted variable bias, but also child level. Of the three approaches, matching children over time (Model 3), despite having only 43% of the non-matched sample size, was the most efficient yielding results with a *p*-value < 0.05 across three variables: WHZ, severe wasting, and underweight.

Across all of our models, girls tended to have better nutrition outcomes compared to boys, which is consistent with other studies on stunting [17,18], underweight, wasting, and under five mortality [18] in the Sahel and southeast Asia [19]. The World Health Organization (WHO) has called on global health institutions to address this gap in both policy and programming as part of the post-2015 SDG agenda [20]. Explanations for this gendered difference, that goes against the expected vulnerability of girls in all other facets of life in the Sahel, range from favoritism for girls [21] to more biological reasons that leave younger boys more vulnerable to mortality and morbidity [22].

There is likely some selection bias associated with the (in)ability to follow-up with the same child over time. Children who can be matched over time are by definition more likely to be present at the household and healthy enough for us to gather anthropometry measurements. However, we found no difference across any of our nutrition indicators at any time period between children that we could and could not match, increasing our confidence in the findings. Future longitudinal studies will need to build in sufficient time and funding to be able to track children even if physically displaced to reduce selection bias. We also recognize that the quality of our age (in months) variable might introduce bias in the analysis. A recent paper done by Larsen et al. (2019) shows how random errors in the month of birth recorded in Demographic and Health Surveys produce nonrandom patterns in HAZ because some children who might have been mistakenly reported as born earlier, appear taller for their age and vice versa [23]. There is a possibility that the same error is present in our data, however this would only affect one of the three indicators for which we observe impact—underweight—because neither WHZ nor severe wasting utilize age in their calculations. The impact on underweight would be similar as on stunting: children who were born earlier than recorded might appear to have a higher weight for their age and vice versa. Given the significance of the other two variables that utilize weight (severe wasting and WHZ), we trust that the impact of the intervention on underweight is accurate, though perhaps the size of the coefficient is biased and should be interpreted with caution.

Overall, while we did not carry out a cost-benefit analysis, despite the smaller sample size, matched child analysis (Model 3) potentially offers a more efficient approach to program impact evaluation as compared to a cross-sectional design (Model 1), due to the ability to control for omitted household and child level characteristics, and thus, yielding more results with a *p*-value < 0.05 despite the smaller sample size. However, the added benefit of panel data collection is highly dependent on the cost of tracking the same children over time. The CRAM study was done in a relatively small geographical area, with no households more than a three-hour drive from the base of operations, and minimal long-term migration of whole households, thus following the same children over time did not add a large additional financial burden.

In addition, the matched household data using the nutritionally worst-off or best-off child in the household as the primary unit of analysis (Model 2) can potentially provide additional insight into whom in the household the program is impacting. We see a large discrepancy in the prevalence of wasting depending on whether we take the nutritionally best-off or worst-off child in the household. For example, at the endline in the control villages, wasting was almost three times higher (27%) if you took the worst-off child compared to the best-off child in the household (10.6%) (Appendix A). Previous studies have identified variability in intra-household resource allocation frequently characterized as “benign” neglect [24]. The authors describe this phenomenon as non-malicious, but rather engendered by pervasive poverty and rooted in the need to balance the perceived risks to children with the preservation of long-term livelihoods. When running the regression on the nutritionally best-off child (Model 2), we see no program impact on WHZ, or any nutrition outcome variable when reducing the sample down to households with two or more children. Thus, using panel household data, but selecting a child within the household based on specific criteria, can offer additional insights about program impact and targeting. Furthermore, this approach can prove useful for panel studies with longer run-times when children might grow out of the inclusion age (for example, under two or five years) or child unique IDs were not collected.

This research also highlights the benefits of collaboration between university and NGO partners. Tufts brought the necessary rigor required to carry out a proper research study by providing input into the design, variable selection, analysis plan, and ethical approvals while also building up the NGOs own monitoring and evaluation capacity. The Concern team provided important contextual know-how and legitimacy to the research. Given the panel nature of data collection and the history of insecurity in eastern Chad, it was important that the respondents trusted the enumerators and saw the value of their time and contribution. This could only be achieved through partnering with an organization that had a history and positive reputation in the community. Furthermore, the Concern team was able to monitor and provide feedback to the respondents in between the periods of data collection given their presence on the ground. Thus, the partnership proved an optimal and cost-effective approach to carrying out rigorous research in a difficult and complex humanitarian environment.

Given the recent interest in multi-sectoral approaches to combat all forms of child malnutrition [4,5,6], this study helps build the evidence base for the efficacy of this approach and offers recommendation for evaluation design. Future policy needs to support programming strategies that tackle the multiple and diverse drivers of child nutritional status. However, if the budget permits, future research would ideally tease out the impact of single focused interventions, such as construction of a well, in comparison with the multi-sectoral approach to show the added value of the combined package as well as tease out context-specific drivers to better tailor programming to its environment.

## 5. Conclusions

This study adds to the small but growing evidence base of the role of multi-sectoral programming on preventing child acute malnutrition. The most robust analysis shows that children who received the full package of activities were 76% less likely to be severely wasted, 33% less likely to be underweight, and had a 0.19 standard deviations higher weight-for-height z-score compared to children in the control group. There was no impact on more long-term outcomes such as height-for-age z-score or stunting.

The analysis presented in this paper goes one step further and tries to better understand the level of sensitivity of different nutrition indicators to three common modeling approaches given how frequently both the outcomes and models are interchanged in the existing literature on multi-sectoral program impact. Despite the large drop in sample size, the findings illustrate that using models that can control for non-measured characteristics at both the household and child level shows more robust program impact in terms of the nutrition indicators tested and the level of significance. Thus, future research, when feasible, should prioritize longitudinal panel studies over a cross-sectional design.

## Figures and Tables

**Figure 1 ijerph-17-03121-f001:**
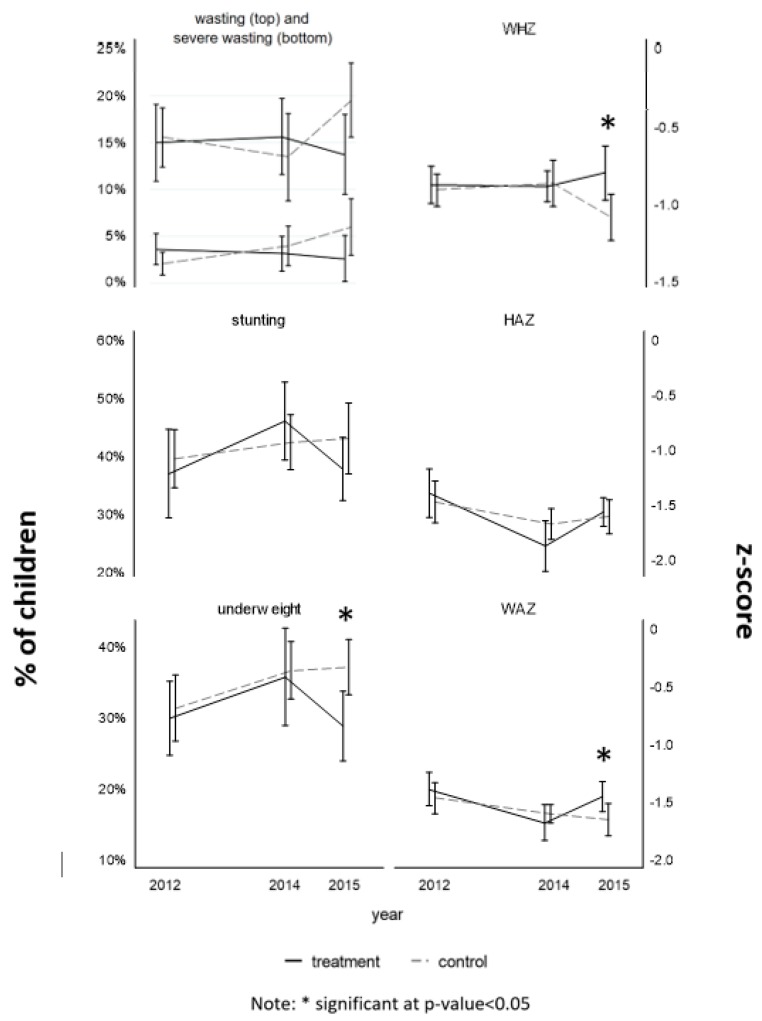
Nutrition outcome indicators (mean and confidence interval) by time and intervention (non-matched child).

**Figure 2 ijerph-17-03121-f002:**
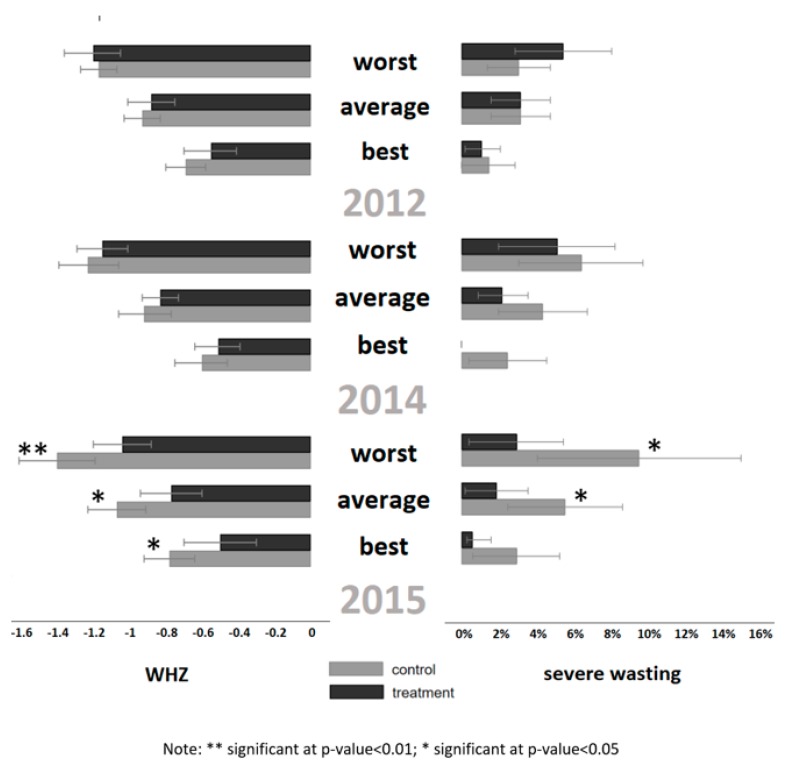
WHZ (left figure) and mean severe wasting (right figure) for nutritionally worst, average, and best-off child in the household (matched household).

**Figure 3 ijerph-17-03121-f003:**
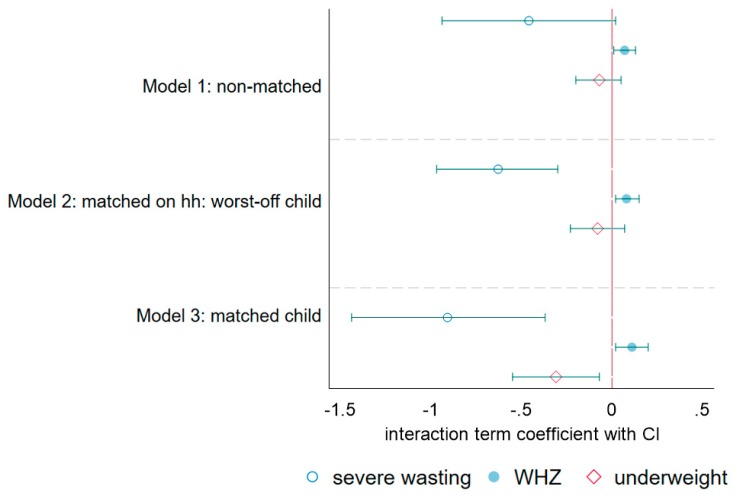
Impact of CRAM on severe wasting, WHZ, and underweight by type of analysis.

**Table 1 ijerph-17-03121-t001:** Variable definition and options.

Variable		Definition	Options
Outcome	WHZ	weight-for-height z-score	−5 to 5 z-score
	Wasting	weight-for-height z-score <−2	1 = wasted 0 = non-wasted
	Severe wasting	weight-for-height z-score <−3	1 = severely wasted 0 = not severely wasted
	HAZ	height-for-age z-score	−6 to 6 z-score
	Stunting	height-for-age z-score <−2	1 = stunted 0 = non-stunted
	WAZ	weight-for-age z-score	−6 to 6 z-score
	Underweight	weight-for-age z-score <−2	1 = underweight 0 = non-underweight
Input	Treatment	whether the hh was in a village randomized to receive CRAM	1 = treatment
			0 = control
	Time	what year the survey was conducted (month = November/December)	0 = 2012
			2 = 2014
			3 = 2015
	Time × treatment	interaction term between time and treatment	0 = control or 2012 2 = treatment in 2014 3 = treatment in 2015
Control	number of children in hh	number of children (in the roster) under the age of 5 in the hh	1 to 5 children
	Female	if the child is female	0 = boy
			1 = girl
	age	age of the child in months	6 to 59 months
	child died in the household	whether the household reported a child 5 years or younger having died in the past year	0 = no
			1 = yes
Design effect	Households	Controlling for household level characteristics	
	Village	Controlling for village level characteristics	

**Table 2 ijerph-17-03121-t002:** Adjusted regression on nutrition outcome indicators by sample ^‡^ (2012, 2014, and 2015).

Sample	Outcome Indicator	Time	Treatment	Time * Treatment	# of Children <5	Female Child	Age (in Months)	Child <5 Died in hh	Constant	n
Non-matched	Wasting	0.07 (−0.03; 0.19)	0.05 (−0.31; 0.42)	−0.09 (−0.27; 0.08)		−0.34 ** (−0.55; −0.12)	−0.00 (−0.01; 0.00))	−0.14 (−0.62; 0.33)	−1.37 *** (−1.75; −0.99)	3462
	Severe wasting	0.36 * (0.03; 0.69)	0.58 (−0.19; 1.37)	−0.45 (−0.93; 0.02)		−0.36 (−0.82; 0.08)	−0.01 (−0.02; 0.00)	−0.26 (−1.26; 0.74)	−3.28 *** (−3.97; −2.58)	3462
	WHZ	−0.04 * (−0.08; −0.00)	−0.01 (−0.16; 0.14)	0.06 * (0.00; 0.12)		0.09 * (0.01; 0.17)	0.00 (−0.00; 0.00)	−0.06 (−0.28; 0.15)	−0.96 *** (−1.11; −0.80)	3462
	Stunting	0.04 (−0.01; 0.10)	−0.03 (−0.39; 0.31)	0.00 (−0.15; 0.13)		−0.24 *** (−0.37; −0.12)	−0.00 (−0.00; 0.00)	0.02 (−0.34; 0.39)	−0.44 * (−0.45; −0.03)	3459
	HAZ	−0.04 (−0.10; 0.00)	−0.02 (−0.24; 0.30)	0.03 (−0.15; 0.07)		0.18 ** (0.07; 0.29)	−0.00 (−0.00; 0.00)	−0.01 (−0.31; 0.28)	−1.57 *** (−1.72; −1.41)	3459
	Under-weight	0.09 ** (0.02; 0.015)	−0.02 (−0.34; 0.28)	−0.07 (−0.20; 0.05)		−0.21 ** (−0.35; −0.07)	−0.00 * (−0.01; −0.00)	0.11 (−0.29; 0.52)	−0.50 *** (−0.78; −0.22)	3522
	WAZ	−0.05 *** (−0.09; −0.02)	0.03 (−0.16; 0.22)	0.01 (−0.05; 0.09)		0.12 *** (0.04; 0.19)	−0.00 (−0.00; 0.00)	−0.03 (−0.26; 0.19)	−1.50 *** (−1.67; −0.13)	3522
Matched on the household (worst-off child)	Wasting	0.03 (−0.09; 0.15)	0.17 (−0.25; 0.61)	−0.13 (−0.31; 0.04)	0.47 *** (0.30; 0.63)	−0.39 ** (−0.63; −0.15)	−0.00 (−0.01; 0.00)	−0.29 (−0.85; 0.25)	−2.13 *** (−2.65; −1.60)	2254
	Severe wasting	0.29 * (0.06; 0.53)	0.86 * (0.19; 1.54)	−0.63 *** (−0.97; −0.29)	0.54 *** (0.27; 0.81)	−0.46 * (−0.88; −0.04)	−0.00 (−0.02; 0.00)	−0.17 (−1.14; 0.80)	−4.36 *** (−5.35; −3.37)	2254
	WHZ	−0.02 (−0.06; 0.02)	−0.06 (−0.25; 0.11)	0.08 ** (0.01; 0.14)	−0.24 *** (−0.30; −0.18)	0.13 ** (0.04; 0.22)	0.00 (−0.00; 0.00)	−0.03 (−0.21; 0.15)	−0.83 *** (−1.02; −0.64)	2254
	Stunting	0.02 (−0.06; 0.12)	−0.23 (−0.56; 0.08)	0.06 (−0.07; 0.21)	0.34 *** (0.20; 0.47)	−0.33 *** (−0.52; −0.14)	−0.00 (−0.00; 0.01)	0.09 (−0.31; 0.49)	−0.67 ** (−1.07; −0.27)	2261
	HAZ	−0.02 (−0.09; 0.04)	0.07 (−0.15; 0.30)	−0.04 (−0.14; 0.05)	−0.25 *** (−0.34; −0.16)	0.23 *** (0.10; 0.36)	−0.00 (−0.01; 0.00)	0.01 (−0.26; 0.28)	−1.41 *** (−1.69; −1.14)	2261
	Under-weight	0.09 (−0.00; 0.20)	0.00 (−0.33; 0.35)	−0.07 (−0.22; 0.06)	0.35 *** (0.21; 0.49)	−0.37 *** (−0.57; −0.16)	−0.00 (−0.00; 0.00)	0.12 (−0.30; 0.54)	−0.87 *** (−1.30; −0.44)	2277
	WAZ	−0.03 (−0.08; 0.01)	0.01 (−0.16; 0.18)	0.02 (−0.04; 0.08)	−0.19 *** (−0.26; −0.13)	0.17 *** (0.08; 0.27)	−0.00 (−0.00; 0.00)	−0.01 (−0.18; 0.20)	−1.42 *** (−1.62; −1.23)	2277
Matched on the child	Wasting	0.06 (−0.13; 0.27)	0.38 (−0.28; 1.06)	−0.24 (−0.52; 0.03)		−0.35 * (−0.69; −0.01)	−0.00 (−0.01; 0.00)	−0.62 (−1.48; 0.24)	−1.87 *** (−2.46; −1.28)	1487
	Severe wasting	0.30 (−0.09; 0.71)	1.28 ** (0.12; 2.44)	−0.90 *** (−1.44; −0.37)		0.20 (−0.36; 0.77)	0.01 (−0.00; 0.03)	−0.39 (−1.84; 1.06)	−4.42 *** (−5.77; −3.08)	1487
	WHZ	-0.04 (−0.10; 0.02)	−0.13 (−0.39; 0.12)	0.10 * (0.01; 0.20)		0.03 (−0.08; 0.15)	0.00 (−0.00; 0.00)	−0.01 (−0.27; 0.25)	−0.92 *** (−1.1-2; 0.71)	1487
	Stunting	0.04 (−0.11; 0.20)	−0.08 (−0.62; 0.45)	−0.06 (−0.15; 0.28)		−0.43 ** (−0.71; −0.15)	−00 (−0.00; 0.01)	−0.09 (−0.72; 0.52)	−0.41 (−0.86; 0.03)	1487
	HAZ	−0.06 (−0.15; 0.03)	−0.11 (−0.43; 0.21)	0.01 (−0.10; 0.14)		0.34 *** (0.18; 0.51)	−0.00 (−0.00; 0.00)	0.26 (−0.08; 0.62)	−1.60 *** (−1.87; −1.34)	1487
	Under-weight	0.20 * (0.02; 0.37)	0.52 (−0.07; 1.12)	−0.30 * (−0.54; −0.06)		−0.48 ** (−0.79; −0.16)	0.00 (−0.00; 0.01)	−0.52 (−1.24; 0.18)	−1.00 *** (−1.51; −0.49)	1505
	WAZ	−0.05 (−0.12; 0.00)	−0.13 (−0.36; 0.09)	0.06 (−0.02; 0.15)		0.19 *** (0.07; 0.31)	−0.00 (−0.00; 0.00)	0.15 (−0.10; 0.40)	−1.47 *** (−1.66; −1.28)	1505

^‡^ coefficient with confidence intervals in parentheses (controlling for population and design effect); * *p* < 0.05; ** *p* < 0.01; *** *p* < 0.001.

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
