# Peer review of "Sensitivity of Nutrition Indicators to Measure the Impact of a Multi-Sectoral Intervention: Cross-Sectional, Household, and Individual Level Analysis"

_ijerph, 2020, doi:10.3390/ijerph17093121_

Round 1

Reviewer 1 Report

The manuscript investigated the sensitivity of nutrition indicators to multi-sectorial interventions as well as different statistical analysis methods. This study is important is pointing out how statistical approach chosen can influence outcome of the study. The following comments are suggested to improve the quality of the manuscript

  1. Line 89-91: There is no explanation why severe wasting was considered in the analysis, while severe underweight and severe stunting were not used.
  2. Line 108-109: Did the authors obtain ethical clearance from Chad? If so, that should be added.
  3. Line 115-116: Authors should justify why only women in the three bottom wealth groups were used for the impact evaluation.
  4. Line 146: The authors have indicated the difficulty in capturing the true age of children. What is the implications of this on the outcomes of the study. Also this need to be mentioned as a limitation
  5. Table 1: Under options, some variables have been ignored. For example, what were the options for severe wasting?
  6. Line 163: Do the authors mean “adapted” instead of adopted?
  7. Figure 1: Please revise figure title. Also only six of the seven malnutrition indicators are shown in the figure.
  8. Figure 3: Please define sam.

Reviewer 2 Report

  1. Why use of GLM and why not multilevel effect.
  2. Second why cant you applied the Diff in diff regression.

Reviewer 3 Report

This was a focused, clear, and well-written examination of various evaluation models for understanding multi-sectoral interventions in childhood malnutrition based on many years of collaborative work.

I am very curious about why girls fared better in GLM and mixed effects models and would have liked more discussion. 

Perhaps the equations (ln 172-203) could be moved to supplemental in order to have more discussion and future recommendations. 
